# Acceleration of Carbon Fixation in Chilling-Sensitive Banana under Mild and Moderate Chilling Stresses

**DOI:** 10.3390/ijms21239326

**Published:** 2020-12-07

**Authors:** Jing Liu, Tomáš Takáč, Ganjun Yi, Houbin Chen, Yingying Wang, Jian Meng, Weina Yuan, Yehuan Tan, Tong Ning, Zhenting He, Jozef Šamaj, Chunxiang Xu

**Affiliations:** 1Department of Pomology, College of Horticulture, South China Agricultural University, Guangzhou 510642, China; 15800231068@163.com (J.L.); hbchen@scau.edu.cn (H.C.); wyy199008@163.com (Y.W.); mengjian2019@163.com (J.M.); 17818521882@163.com (W.Y.); t13416479181@163.com (Y.T.); nt8023@163.com (T.N.); 18734453308@163.com (Z.H.); 2Centre of the Region Haná for Biotechnological and Agricultural Research, Department of Cell Biology, Faculty of Science, Palacký University, 783 71 Olomouc, Czech Republic; tomas.takac@upol.cz (T.T.); jozef.samaj@upol.cz (J.Š.); 3Guangdong Province Key Laboratory of Tropical and Subtropical Fruit Tree Research, Institute of Fruit Tree Research, Guangdong Academy of Agricultural Sciences, Guangzhou 510640, China; yiganjun@vip.163.com

**Keywords:** banana (*Musa* spp. AAA), mild chilling, carbon fixation, photosynthesis, phosphoenolpyruvate carboxylases, immunofluorescence labeling

## Abstract

Banana is one of the most important food and fruit crops in the world and its growth is ceasing at 10–17 °C. However, the mechanisms determining the tolerance of banana to mild (>15 °C) and moderate chilling (10–15 °C) are elusive. Furthermore, the biochemical controls over the photosynthesis in tropical plant species at low temperatures above 10 °C is not well understood. The purpose of this research was to reveal the response of chilling-sensitive banana to mild (16 °C) and moderate chilling stress (10 °C) at the molecular (transcripts, proteins) and physiological levels. The results showed different transcriptome responses between mild and moderate chilling stresses, especially in pathways of plant hormone signal transduction, ABC transporters, ubiquinone, and other terpenoid-quinone biosynthesis. Interestingly, functions related to carbon fixation were assigned preferentially to upregulated genes/proteins, while photosynthesis and photosynthesis-antenna proteins were downregulated at 10 °C, as revealed by both digital gene expression and proteomic analysis. These results were confirmed by qPCR and immunofluorescence labeling methods. Conclusion: Banana responded to the mild chilling stress dramatically at the molecular level. To compensate for the decreased photosynthesis efficiency caused by mild and moderate chilling stresses, banana accelerated its carbon fixation, mainly through upregulation of phosphoenolpyruvate carboxylases.

## 1. Introduction

Low temperature (LT) is one of the major environmental factors which not only limit the productivity and the geographical distribution of crops but also cause significant losses [1,2]. In recent years, modern high-throughput expression profiling techniques, such as genomics, transcriptomics, and proteomics, have provided useful tools for better understanding the roles of genes when responding to LT stresses [3,4,5,6,7]. However, most of these studies reported about the effects of severe LT (≤10 °C) on temperate [5,7,8,9], tropical, and subtropical [3,4,10] plants. Thus, information about the gene expression dynamics of tropical crops in response to mild (>15 °C) and moderate chilling (10–15 °C) is scarce [11,12], and the underlying molecular background is not well understood.

Banana (*Musa* spp.) is a very important tropical crop with annual production of 153 million tons [13] and is very sensitive to LT [1]. Banana grows best when the temperature is between 24 and 32 °C, while its growth ceases at 10–17 °C [14], depending on the species and cultivars, developmental stage, and the length of exposure to LT [15]. The critical LT for the growth of most cultivated banana cultivars in China is approximately 13 °C [16]. Chilling temperatures that range between 10 and 15.5 °C frequently appeared in winter and early spring in tropical and subtropical banana production regions and seriously threaten the productivity of banana [17]. Therefore, it is very necessary to understand the molecular mechanism of banana tolerance to mild to moderate chilling.

High-throughput techniques have also been employed to reveal the mechanism of banana tolerance to LT stresses. A proteomic analysis was carried out with a chilling-tolerant (CT) cultivar (*Musa* spp. ABB) after exposure to 8 °C [10] and a chilling-sensitive (CS) one treated at 5 °C [18]. A total of 809 differentially expressed genes (DEGs) showed differential expression in the CT banana cultivar, and many of these genes were involved in oxidation–reduction reactions, photosynthesis, and photorespiration [10]. However, only 41 protein spots exhibited a change in intensity by at least 2-fold in the CS one, including several proteins related to photosynthesis [18]. Yang et al. [3] compared different molecular responses of a CS *Musa* spp. AAA and CT *Musa* spp. ABB to LT of 10 °C (LT10) using RNA sequencing (RNA-Seq). The involved pathway related to the 17 DEGs uniquely present in the CT cultivar included abiotic stress, signal transduction, photosynthesis, and photorespiration, and so on. Recently, changes in mRNA and long noncoding RNAs (lncRNAs) of a CT wild banana (*Musa itinerans*) with growth ceasing temperature at 0 °C [19] in response to different LTs (13, 4, and 0 °C) were analyzed, and photosynthesis, photosynthetic-antenna proteins, circadian rhythm, starch and sucrose metabolism, and cutin and suberine biosynthesis were found to be altered by these cold stress conditions [20]. Thus far, to our knowledge, there is no study focused on the molecular responses of the CS banana cultivar to LT above 10 °C, a stress frequently encountered by banana and other tropical crops.

Photosynthesis is one of the most temperature-sensitive processes in plants. To be adapted to the fluctuation in temperature, most plants could adjust their photosynthetic characteristics considerably [21]. Limited information exists about the photosynthesis acclimation strategies of banana during mild LT stress.

Therefore, here we aimed to reveal the whole-transcriptome gene expression patterns of the CS banana cultivar “Baxijiao” (*Musa* AAA, with annual production of several million tons) to moderate LT10 and mild LT of 16 °C (LT16). We complemented this transcriptome with proteome changes in response to LT10. By this approach, we identified the transcriptional and protein landscape associated with its response of banana to mild to moderate chilling. In addition, we focused on the photosynthesis responses of this banana cultivar to LT stresses at the molecular (transcripts, proteins) and physiological levels. The results suggest that CS banana responded to LT16 more strongly at the molecular level when compared to LT10, and there were considerable differences in gene expression profiles between LT16 and LT10. Most importantly, our data revealed that the expression and abundance of genes/proteins related to carbon fixation and the majority of key photosynthetic enzymes increased significantly in CS banana after exposure to LT10 and/or LT16.

## 2. Results

### 2.1. Morphological and Physiological Changes of Banana in Response to Mild and Moderate Chilling Stress

The banana seedlings started to show light wilting due to water deprivation at 16 °C after 2 days, as shown in Figure 1a,b. Hydrophanous spots could be observed on the youngest fully developed leaf 2 days after the temperature drop to 10 °C, as shown in Figure 1c. The relative electrolyte leakage of the control (CK, at 25 °C) banana plants was 22.53%. It significantly increased to 26.15% when the temperature dropped to 16 °C and further increased to 28.00% at 10 °C, but there was no significant difference between LT10 and LT16. Similarly, the leaf damage rate also continuously increased with the temperature drop, as shown in Figure 1d. These data show that both LT16 and LT10 had damaging effects on membranes in banana. The Rubisco activity of banana increased after exposure to LT, though not at significant level, as shown in Figure 1e, pointing to the preservation of carboxylation processes in the CS banana cultivar. This is consistent with the stabile intercellular CO_2_ concentration under both treatments, as shown in Figure 1f. In addition, the phosphoenolpyruvate carboxylase (PEPC) activity significantly increased from 0.64 μmol g^−1^ min^−1^ (CK) to 1.27 μmol g^−1^ min^−1^ at LT10, though there was a slight decrease when the temperature dropped from 25 °C to 16 °C, as shown in Figure 1g. The stomatal conductance was substantially affected by both temperatures and it decreased significantly from 251.8 mmol m^−2^ s^−1^ at 25 °C to approximately 85 mmol m^−2^ s^−1^ after the exposure to LT, as shown in Figure 1 h. A slight decline was observed with both the soil plant analysis development (SPAD) value (indicating the relative content of chlorophyll), as shown in Figure 1i, and the net photosynthetic rate in LT treated banana plants, as shown in Figure 1j. Differently, the quantum efficiency of photosystem II (Fv/Fm) in banana decreased significantly from 0.75 to 0.51 when the temperature dropped to 10 °C, as shown in Figure 1k, but it was unaffected at 16 °C.

### 2.2. Transcriptomic Analysis of Banana Exposed to the Mild and Moderate Chilling Stress

The RNA-Seq (digital gene expression, DGE) technique was employed to reveal the responses of a CS banana cultivar to mild and moderate chilling at transcript level. In total, six cDNA libraries were constructed consisting of two biological replicates each of CK, plants exposed to LT16 and to LT10, and subjected to sequencing on an Illumina HiSeq 2000 platform.

For easier analysis of results, DEGs, and the Kyoto Encyclopedia of Genes and Genomes (KEGG) among LT16 and CK, LT10 and CK, and LT10 and LT16 were designated as LT16-CK, LT10-CK, and LT10-LT16 in the present study. **|**log2 (fold change)**|** > 1 and a false discovery rate (FDR) < 0.05 were used as the threshold to evaluate the significance of DEGs.

#### 2.2.1. Transcriptomic Analysis of Banana Exposed to 16 °C Chilling Stress

Treatment of banana with LT16 lead to the identification of 1933 DEGs, out of which 978 were upregulated and 955 were downregulated, as shown in Appendix A. All DEGs induced by LT16 were classified into 105 KEGG pathways, while the upregulated and downregulated genes were annotated by 82 and 85 annotations, respectively, as shown in Appendix A. The top 20 KEGG pathways (with the smallest FDR values) related to upregulated genes are shown in Appendix A. These mainly included carbon fixation, flavonoid biosynthesis, circadian rhythm, carbon metabolism, and cutin, suberine, and wax biosynthesis. Annotations related to circadian rhythm, plant-pathogen interaction, starch and sucrose metabolism, galactose metabolism, tropane, piperidine, and pyridine alkaloid biosynthesis ranked within the first five ones for the downregulated DEGs, as shown in Appendix A.

#### 2.2.2. Transcriptomic Analysis of Banana Exposed to 10 °C Chilling Stress

In total, LT10 induced differential expression of 1267 DEGs, comprising 724 upregulated and 543 downregulated DEGs, as shown in Appendix A.

DEGs induced by LT10 were classified into 103 KEGG pathways, including 89 and 72 pathways connected with up- or downregulated genes, respectively, as shown in Appendix A. The top 20 KEGG pathways related to upregulated genes are shown in Appendix A. The first five pathways (according to the corrected *p* value) were glycolysis/gluconeogenesis, biosynthesis of secondary metabolites, carbon fixation, carbon metabolism, and biosynthesis of amino acids. Downregulated genes were mainly assigned to galactose metabolism, photosynthesis, photosynthesis-antenna proteins, starch and sucrose metabolism, and cyanoamino acid metabolism, as shown in Appendix A.

#### 2.2.3. Comparison of LT10 and LT16

The comparison of transcriptomic data obtained at LT10 and LT16 provided 279 DEGs, while 175 of them were upregulated and 104 were downregulated, as shown in Appendix A.

DEGs of the LT10–LT16 comparison were classified into 36 KEGG pathways, as shown in Appendix A, including 24 and 18 pathways connected with up- or downregulated genes, respectively. The top 20 KEGG pathways related to upregulated genes included plant-pathogen interaction, glutathione metabolism, arachidonic acid metabolism, selenocompound metabolism, and starch and sucrose metabolism, as shown in Appendix A. KEGG pathways related to downregulated genes are assigned to plant hormone signal transduction, ABC transporters, ubiquinone and other terpenoid-quinone biosynthesis, as well as starch and sucrose metabolism, and cutin, suberine, and wax biosynthesis, as shown in Appendix A.

#### 2.2.4. Evaluation of KEGG Pathways Enrichment

In total, when taking the corrected *p* value ≤ 0.05 as the threshold, 12 KEGG pathways showed significantly differentially expressed enrichment. KEGG pathways analysis resulted in five significantly enriched items (carbon fixation in photosynthetic organisms, flavonoid biosynthesis, circadian rhythm-plant, plant-pathogen interaction, starch and sucrose metabolism) for DEGs found in response to LT16. Another four KEGG pathways (glycolysis/gluconeogenesis, biosynthesis of secondary metabolites, carbon metabolism, and galactose metabolism) significantly enriched after LT10 except carbon fixation in photosynthetic organisms, which were also present in LT16. There were three significantly differentially expressed KEGG pathways between these two LT treatments, namely plant hormone signal transduction, ABC transporters, and ubiquinone and other terpenoid-quinone biosynthesis, as shown in Figure 2.

### 2.3. Proteomic Analysis of Banana Exposed to Mild Chilling Stress

For better understanding of banana response to moderate chilling at the protein level, the changes of proteome of CS banana cultivar leaves treated by 10 °C were analyzed by LC-MSMS combined with tandem mass tag quantification. In total, 110,693 out of 429,482 mass spectrum graphs and 5744 proteins were identified in the present study. Analysis revealed 366 differentially abundant proteins after LT10 treatment, out of which 196 proteins exhibited increased and 170 decreased abundance, as shown in Appendix A.

Data were evaluated by KEGG pathway analysis. These proteins could be assigned to 76 KEGG pathways, including the 10 most enriched ones with an FDR value < 0.0005. These included carbon fixation in photosynthetic organisms, photosynthesis-antenna proteins, photosynthesis, porphyrin and chlorophyll metabolism, carbon metabolism, as well as metabolic pathways and biosynthesis of secondary metabolites, as shown in Figure 3.

### 2.4. DEGs and Proteins Involved in Photosynthesis upon LT Stress

#### 2.4.1. DGE Analysis

Photosynthesis is one of the most LT-sensitive processes in plants. As mentioned above, carbon fixation in photosynthetic organisms was the only KEGG pathway significantly enriched commonly at both LT16 and LT10. Thus, we constructed a heat map for 26 and 19 DEGs related to carbon fixation and photosynthesis (FDR < 0.05), respectively, as shown in Figure 4. This showed that except for two genes (pyruvate, phosphate dikinase, chloroplastic; and ribulose bisphosphate carboxylase small chain, chloroplastic), the other 24 DEGs related to carbon fixation showed relatively higher expression levels at LT16 and LT10 than at 25 °C, as shown in Figure 4a. On the other hand, 11 (3 gene encoding proteins of PSI, 4 gene encoding proteins of PSII, and 4 genes assigned to KEGG pathway antenna proteins) out of 19 DEGs involved in light reactions of photosynthesis were downregulated in response to both LT16 and LT10, as shown in Figure 4b. On the other hand, six (chlorophyll a–b binding protein CP26, chlorophyll a–b binding protein 21, ferredoxin-2, ferredoxin--NADP reductase, cytochrome b6-f complex iron-sulfur subunit, and the ATP synthase F1-dtta subunit family) of them were upregulated by LT16. However, their expression levels decreased at 10 °C but remained higher than at 25 °C. Only one gene (photosystem II 22 kilodalton (kDa) protein, chloroplastic) was upregulated at both LT16 and LT10 when compared to the control. The MapMan about DEGs involved in photosynthesis at LT16 and LT10 are shown in Appendix A, respectively. The gene list identified in the present study is provided in the Appendix A.

#### 2.4.2. Proteomics Analysis

The proteomic analysis showed that 12 proteins involved in PSI, PSII, and light-harvesting chlorophyll II protein complex/photosynthesis (antenna proteins) were downregulated. In contrast, 10 proteins related to carbon fixation in photosynthetic organisms were upregulated under LT, as shown in Figure 5 and Figure 6. A complete list of differentially regulated proteins including details of their quantification is provided in the supporting information, as shown in Appendix A.

As mentioned above, proteins and genes related to carbon fixation were upregulated in banana leaves in response to mild and moderate LT stress. Eight proteins/genes showed the same trends in both proteomics and RNA-Seq analyses. Among these, phosphoenolpyruvate carboxylase-housekeeping isozyme, glyceraldehyde-3-phosphate dehydrogenase A, and alanine aminotransferase 2 were upregulated at both the transcript and protein levels, while chlorophyll a-b binding protein 6A, oxygen-evolving enhancer protein 2, photosystem II 10 kDa polypeptide, photosystem II 11 kDa protein, and a hypothetical protein were downregulated. Another six proteins also showed consistent trends with their corresponding genes upon LT stress, although their changes at the transcript level were not significant due to the q value (the expected number of fragments per kilobase of transcript sequence per million base pairs sequenced; value difference was as high as or greater than 2 times). Among these six genes/proteins, three (photosystem I reaction center subunit N, chlorophyll a-b binding protein CP26, and putative chlorophyll a-b binding protein type 1 member F3) were downregulated while the other three (lactate/malate dehydrogenase, glyceraldehyde-3-phosphate dehydrogenase B, and phosphoglycerate kinase) were upregulated.

### 2.5. Validation of Transcriptomic Data Using qPCR Analysis

PEPC, Rubisco and pyruvate, phosphate dikinase (PPDK) are the key enzymes responsible for the photosynthesis of monocotyledons. Interestingly, nearly all members of these three families responded to LT significantly except three ones (*MarbcS1*, *MaRca1*, *MaPEPC6*), suggesting the importance of these three gene families in banana under mild and moderate LT stress. To validate these changes, the expression levels of all 20 genes belonging to these gene families before and after LT treatments were analyzed by qPCR. Among the six members of the PEPC family, the expression of *MaPEPC2* increased significantly and the other four isoforms also showed insignificant increase in expression in response to LT; *MaPEPC6* showed reduced levels, as shown in Figure 7a–f. Rubisco activase *MaRca2* showed significant upregulation after LT10 but not LT16, as shown in Figure 7g–o. On the other hand, gene encoding of a small subunit of Rubisco, *MarbcS4,* was downregulated significantly by both LT16 and LT10. Though no significant differences were observed with the expression levels of the other seven Rubisco subunit isoforms in LT-treated banana, two of them are showing a slight increase while another four are regulated oppositely, as shown in Figure 7g–o. Similarly, two PPDK members showed a slight increase in expression, while another two isoforms were reduced, as shown in Figure 7p–t. In addition, we also validated the expression of randomly selected 21 DEGs involved in photosynthesis. The changes in their expression levels were consistent with those obtained from DGE analysis. Eight of the twelve genes assigned to carbon fixation pathway (*MaMDH2*, *MaMDH3*, *MaGAPCp1*, *MaGAPA1*, *MaGAPB1*, *MaNADP-ME1*, *MaNADP-ME2*, *MaNADP-ME3*) were upregulated, as shown in Figure 8a–l. Meanwhile, nine genes associated to photosynthesis and photosynthesis antenna were downregulated by LT, as shown in Figure 8m–u.

### 2.6. Immunofluorescence Labeling of Photosynthetic Proteins in Banana during Chilling Stress

To confirm the LT-induced changes of some genes/proteins involved in photosynthesis and carbon fixation, we employed immunofluorescence labeling using three antibodies specifically recognizing the key enzymes involved in carbon fixation and two proteins of photosystems. Antibodies recognizing the PSI type I chlorophyll a/b-binding protein (AS1005) and the 23 kDa protein of the oxygen evolving complex of PSII (PsbP) (assigned as AS06142.23) showed a strong signal in the mesophyll and guard cells, as shown in Figure 9a,b. Immunolabeling using antibodies recognizing the Rubisco small subunit (assigned as AS07259) and PEPC (assigned as AS09458) were also localized in the mesophyll and guard cells, but showed much weaker signals when compared to AS1005 and AS06142.23, especially in the mesophyll cells, as shown in Figure 9c,d. The Rubisco large subunit (immunolabeled with antibody assigned as AS03037) appeared in all cross sections through the banana leaves, providing a strong signal in the “Kranz” leaf vein and mesophyll cells, as shown in Figure 9e. As shown in Figure 6f, when the temperature dropped to 16 °C and 10 °C, the antigens of AS1005 and AS07259 were significantly upregulated. When the temperature further dropped to 10 °C, the expression levels of the Rubisco large subunit and PEPC were also significantly higher than those in CK. No significant changes were observed for the oxygen evolving complex, as shown in Figure 9f.

## 3. Discussion

### 3.1. Mild and Moderate Chilling Results in Substantial Changes of Gene/Protein Expression in Banana

Mild and moderate chilling stresses are frequently encountered by banana and other tropical crops. However, nearly all previous studies on banana chilling tolerance were only related to LT ≤ 10 °C [3,10,18].

Omics technologies have brought about a significant breakthrough in the knowledge about plant responses to LTs [3,4,5,6,7]. In the present study, global differences in mRNAs caused by mild and moderate chilling were systematically identified and analyzed in China’s most important banana cultivar with an annual yield of several million tons. The results showed that this banana cultivar responded to LT16 more dramatically, and differently, compared to LT10. In total, 1933 DEGs were identified in banana leaves after exposure to 16 °C with comparison to 1267 ones at 10 °C, while it was 279 for LT10–LT16. The most significantly differentially expressed KEGG pathways between these two LT points were related to plant hormone signal transduction, ABC transporters, ubiquinone, and other terpenoid-quinone biosynthesis. On the other hand, many annotations were overabundant between LT16 and LT10. Upon stress of LT ≥ 10 °C, this CS banana cultivar accelerated carbon fixation, carbon metabolism, upregulated biosynthesis of flavonoid, cutin, suberine, and wax, but downregulated photosynthetic proteins related to antenna complex.

Similarly, a wild banana, which is extremely cold resistant and resists temperatures close to 0 °C [19], was used to profile the cold-responsive mRNAs and lncRNAs by RNA-seq at different LT points (13, 4, and 0 °C) [20]. In total, 1530 transcripts were found to be differentially expressed at 13 °C, and the most enriched 20 KEGG pathways also included photosynthesis-antenna proteins and carbon fixation. However, whether these pathways were up or downregulated is not mentioned. In CS maize (*Zea mays*), however, few photosynthesis-related genes were repressed by chilling, though hundreds of transcripts were affected by LT of 14 °C [11]. In contrast, *Miscanthus* × *giganteus*, an exceptional CT C_4_ species closely related to maize, could keep its ability to acclimate to a chilling condition (≤14 °C) and maintain relatively high photosynthetic capacity. This was reflected also in transcript profiles of photosynthetic proteins, because genes associated with photosynthetic light reactions were upregulated, while they were reduced in CS maize after 14 days of treatment at 14 °C [12]. Thus, the response of tropical banana to LT ≥ 10 °C showed partial similarity to CS crop maize, suggesting that the responses of tropical plants to LT ≥ 10 °C exhibit certain specificities.

### 3.2. Upregulation of Carbon Fixation and Downregulation of the Electron Transport Chain in the Banana Response to Mild and Moderate Chilling Stresses

Photosynthesis is the most fundamental and intricate physiological process in all green plants. Abiotic stresses, including LT, seriously affect plant photosynthesis through disruption of the components of photosynthesis system, including chlorophyll photosystem I and II, electron transport, as well as carbon fixation, and stomatal conductance [22]. Many previous studies have demonstrated that LT stress inhibits photosynthesis, as indicated by reduction in the net photosynthetic rate, stomatal conductance, and maximum photochemical efficiency (Fv/Fm) [10,23,24].

To survive in a fluctuating environment, plants develop some mechanisms to adapt to the new conditions, such as adjustment of cell wall metabolism and adaptation to photosynthetic machinery [25]. In this study, both RNA-seq and proteomics results indicated that light reactions were downregulated in banana upon mild chilling stress. However, mild to moderated LTs upregulated carbon fixation and did not affect the intercellular CO_2_ concentration, though the stomatal conductance was significantly decreased. Such conditions supported the elevated carboxylation reactions (Rubisco, PEPC, and PPDK) as found by biochemical, proteomic, transcriptomic, and immunolabeling analyses. An increase in protein levels of the Rubisco large subunit was also observed in a CT banana genotype (*Musa* spp. ABB) after exposure to 10 °C [10] and a CS banana genotype (*Musa* spp. AAA) after exposure to 5 °C [18]. These results suggested that banana maintained the intracellular CO_2_ concentration through upregulation of PEPCs and PPDKs, which catalyze the regeneration of PEP under mild and moderate chilling stresses. Such increased PEPC levels might enhance the carbon assimilation rate and thereafter support the acclimation to LT in order to maintain the net photosynthetic rate upon LT ≥ 10 °C, although many genes related to PSII and photosynthesis-antenna proteins were downregulated. Similar to our results, the protein levels of Rubisco activase and the expression of genes of the Benson–Calvin cycle simultaneously increased also in CS watermelon (*Citrullus lanatus* (Thunb.) Matsum and Nakai) in response to LT [26]. Naidu et al. [27] revealed that photosynthesis, PPDK, and the Rubisco large subunit decreased by 80%, 50%, and 30%, respectively, in CS maize during LT, whereas these levels remained unaffected in its close C_4_ plant, the CT *Miscanthus* × *giganteus*. The authors suggested that the maintenance of PPDK and Rubisco large subunit expression in *M*. *giganteus* were critical for high rates of C_4_ photosynthesis at LT. Similar results were obtained by more recent observations for maize, *M.* × *giganteus*, and related species during LT stress [11,12,23,28,29,30,31]. These results suggested that CT *M*. × *giganteus* under chilling stress maintains Asat and photosynthesis through keeping stable or even increasing levels and activities of PPDK and Rubisco, as well as genes/enzymes associated with chloroplast membrane function. On the other hand, LTs resulted in a significant decrease of Asat and photosynthesis in CS maize through the loss of efficiency of PSII, which is associated with impairment in the synthesis of key PSII and light-harvesting complex proteins [32]. Differently, it has been proven that the activity and activation of PEPC and Rubisco were not significantly affected by chilling in *Paspalum dilatatum* [33], and the chilling-dependent inhibition of photosynthesis was not related to the maximal phosphoenolpyruvate carboxylation rate [34]. In the present study, no significant change in SPAD was observed in chilling treated banana, but it was not the case for rice as a model C_3_ plant [35]. These results suggest that the photosynthetic system of plants reacts to the stress differently, according to the plant type, photosynthetic system (C_3_ or C_4_), type of stress, time and duration of the stress occurrence, and several other factors [25].

## 4. Materials and Methods

### 4.1. Plant Materials and LT Treatment

A CS banana cultivar “Baxijiao” (*Musa* AAA, with annual production of several million tons) was used as the plant material. Plant preparation and LT treatment were carried out as described by Yan et al. [36]. The samples for proteomic analysis were collected 2 days after being treated at 25 °C and 10 °C for two days, respectively, while it was 25, 16, and 10 °C for all the other analyses.

### 4.2. Measurement of Physiological Parameters

All physiological analyses were performed on laminas of the second fully developed leaf in three biological replicates. One plant was analyzed within each biological replicate.

The electric conductivity was tested according to the method described by Meng et al. [37].

Relative chlorophyll content was measured nondestructively with a SPAD (Minolta, Osaka, Japan) 502 Chl Meter.

The intercellular CO_2_ concentration (Ci), the stomatal conductance, and the net photosynthetic rate were measured under standardized conditions with a portable photosynthesis system (PP system TPS-2, Amesbury, MA, USA). The measurement conditions were: 70% relative air humidity and 420 μmol mol^−1^ ambient CO_2_ concentration. Three plants were taken as a replicate from each treatment.

#### 4.2.1. Chlorophyll Fluorescence Imaging

Stress tolerance was measured based on changes in the maximal photochemical efficiency (Fv/Fm) after LT treatment. Fv/Fm was determined with imaging PAM (IMAG-MAXI; Heinz Walz, Effeltrich, Germany) after the plants were dark adapted for 30 min. Fv/Fm was determined in the whole leaf [38].

#### 4.2.2. Determination of Enzyme Activities of PEPC and Rubisco

The activity of PEPC and Rubisco was measured as described by Wang et al. [39].

For the determination of PEPC, leaf tissue (about 0.2 g) was homogenized in 1.5 mL extraction buffer containing 0.1 M Tris-H_2_SO_4_ (pH 8.2), 1 mM ethylenediaminetetraacetic acid, 7 mM 2-mercapoethanol, 5% (*v*/*v*) glycerol, and 3% (*w*/*v*) insoluble polyvinylpolypyrrolidone. The homogenate was centrifuged at 16,000× *g* at 4 °C for 10 min. Following addition of 0.243 g ammonium sulfate to 1.0 mL supernatant, it was kept at 4 °C for 2 h. Afterwards, the mixture was centrifuged at 16,000× *g* for 15 min. The precipitate was resuspended in 0.2 mL extraction buffer. PEPC activity was measured spectrophotometrically by following the reduction of nicotinamide adenine dinucleotide (NADH) at 340 nm in 1 mL of mixture containing 50 mM Tris-HCl (pH 9.2), 4 mM phosphoenolpyruvate, 10 mM Mg_2_SO_4_, 10 mM NaHCO_3_, 0.1 mg NADH, and 5U malate dehydrogenase. The enzyme activity was calculated from the absorbance value.

For the determination of Rubisco, in brief, frozen leaf discs (about 0.2 g) were ground with a pre-cooled mortar and pestle in 1.5 mL extraction buffer containing 50 mM Hepes-KOH (pH7.5), 10 mM MgCl_2_, 2 mM ethylenediaminetetraacetic acid, 10 mM dithiothreitol, 1% (*v*/*v*) Triton X-100, 5% (*w*/*v*) insoluble polyvinylpolypyrrolidone, 1% (*w*/*v*) bovine serum albumin, and 10% (*v*/*v*) glycerol. The extract was centrifuged at 13,000× *g* for 5 min in an Eppendorf microcentrifuge, and the supernatant was used immediately for an enzyme activity assay. Rubisco activity was determined at 340 nm in a mixture (1 mL) of 100 mM bicine pH 8.0, 25 mM KHCO_3_, 20 mM MgCl_2_, 3.5 mM ATP, 5 mM phosphocreatine, 5 units glyceraldehyde-3-phosphate dehydrogenase, 5 units 3-phosphoglyceric phosphokinase, 17.5 units creatine phosphokinase, 0.25 mM NADH, and 0.5 mM ribulose-1,5-bisphosphate. Sample (20 µL) and ribulose-1,5-bisphosphate were added prior to measurement followed by thorough mixing. The absorbance was monitored at 340 nm.

### 4.3. Immuno-labeling of Antibodies Related to Photosynthesis

Fixation and embedding of samples were carried out according to Xu et al. [40] with the following modifications—an additional step was added between the rinse with phosphate-buffered saline (PBS) after dewaxing and blockage in PBS supplemented with 50 mM glycine. The sections were incubated in 0.1 M citrate buffer (pH 6.0) at 100 °C for 15 min for retrieval of the antigens. The primary antibodies included anti-RbcL (Rubisco large subunit; AS03037), anti-RbcS (Rubisco small subunit; AS07259), anti-Lhca1 (PSI type I chlorophyll a/b-binding protein; AS01005), anti-PsbP (23 kDa protein of the oxygen evolving complex of PSII; AS06142), and anti-PEPC (PEPC; AS09458), diluted in PBS containing 1% (*w*/*v*) bovine serum albumin to a final concentration of 0.5 µg/100 µL. The secondary antibody was goat anti-rabbit IgG (H&L) (AS09634). All of these antibodies were from Agrisera (Vännäs, Sweden). Sections probed only with the secondary antibodies were used as negative controls. Three biological replicates were prepared for each treatment. An Axio Imager D2 (ZEISS, Oberkochen, Germany) used to examine the fluorescence.

### 4.4. DGE Analysis

RNA preparation, library preparation for DGE sequencing, and data analysis were carried out as described previously [41]. FDR was used to determine the threshold of the *p* value in multiple tests and analyses. In the present study, **|**log2 (fold change)**|** > 1 and a threshold of FDR values < 0.05 were used as the threshold to evaluate the significance of DEGs.

### 4.5. Proteomic Analysis

#### 4.5.1. Sample Preparation and Protein Digestion

The samples were ground in liquid nitrogen and extracted in lysis buffer (8 M urea, 2% SDS, 1× protease inhibitor cocktail). Then, the samples were sonicated on ice and centrifuged at 13,000 rpm for 10 min at 4 °C. The supernatant was precipitated with chilled acetone at −20 °C overnight, followed by three times of washing with 50% ethanol and 50% acetone. The protein concentration was determined with the BCA protein assay and 100 μg of total protein was redissolved in 200 μL 8 M urea in 0.1 M Tris/HCl (pH 8.5). Digestion of proteins was performed using the filter aided sample preparation method and was then performed as previously described [42].

#### 4.5.2. Tandem Mass Tags Labeling and High-pH Reversed Phase Separation

The released peptides were collected, lyophilized by SpeedVac, and then labeled with TMT 6-plex reagent according to the manufacturer’s instructions. All of the labeled samples were pooled and dried by vacuum centrifugation, followed by separation with high-pH RP chromatography. In detail, the mixed peptide sample was redissolved in buffer A (20 mM ammonium formate in water, pH 10.0) and gradient-eluted in buffer B (20 mM ammonium formate in 80% ACN, pH 10.0) from 5% to 45% B over 40 min at a flow rate of 1 mL/min in the column (XBridge C18, 4.6 × 250 mm, 5 μm, Waters Corporation, Milford, MA, USA) on an Ultimate 3000 system (Thermo Fisher Scientific, Waltham, MA, USA). Twelve fractions were collected and lyophilized by SpeedVac.

#### 4.5.3. LC-MS/MS Analysis

The peptides were redissolved in solvent C (C: 0.1% formic acid in water) and analyzed on an EASY-nLC 1000 system connected to an Orbitrap Fusion Tribrid mass spectrometer (Thermo Fisher Scientific, Waltham, MA, USA). The peptide sample was loaded (trap column (Thermo Fisher Scientific Acclaim PepMap C18, 100 μm × 2 cm), analytical column (Acclaim PepMap C18, 75 μm × 15 cm)) and separated with a 120 min gradient from 3% to 32% D (D: 0.1% formic acid in ACN). The mass spectrometer was run under a data dependent acquisition mode and automatically switched between MS and MS/MS mode. The parameters were: (1) MS: scan range (*m*/*z*) = 350–1550; resolution = 120,000; (2) high energy collisional dissociation MS/MS: resolution = 30,000; isolation window = 1.6; automatic gain control target = 4 × 10^5^; collision energy = 30.

#### 4.5.4. Protein Identification and Quantification

All MS/MS spectra were searched against the UniProt *Musa malaccensis* proteome database using Mascot (v2.5.1) with the following parameters: peptide tolerance, 7.0 ppm; MS/MS tolerance, 0.05 Da; tryptic digestion; cysteine carbamidomethylation and TMT6plex of lysine and peptide N terminus as fixed modifications; oxidation on methionine, acetylation on the protein N terminus, and deamidation of asparagine and glutamine as variable modifications.

Scaffold (v4.7.5, Proteome Software Inc., Portland, OR, USA) was used to validate the MS/MS-based peptide and protein identification and quantification. Peptide identifications were accepted if they could achieve an FDR less than 1%, while protein identifications were accepted if they contained at least two independent peptides.

Normalization was performed iteratively (across samples and spectra) on intensities as described [43]. Medians were used for averaging. Spectra data were log-transformed, pruned of those matched to multiple proteins and those missing a reference value, and weighted by an adaptive intensity weighting algorithm. Differentially expressed proteins were determined by applying a Mann–Whitney test corrected by Benjamini–Hochberg with significance level *p* < 0.02060 and fold change over 1.3.

#### 4.5.5. Bioinformatic Analysis

The bioinformatic analyses (partial least squares discrimination analysis, volcano plot, hierarchical clustering analysis, gene ontology, KEGG, protein–protein interaction) were performed by using an on-line platform called iOmics Cloud (https://iomicscloud.thermofisher.cn/#!/login).

### 4.6. qPCR

The qPCR analysis was carried out according to the method described in Meng et al. [36]. The primers used are listed in Appendix A.

### 4.7. Statistical Analysis

Statistical analyses were performed using analysis of variance (ANOVA) by using the statistical program SPSS 19.0 for Windows (SPSS Inc., Chicago, IL, USA). Three replicates were set for each treatment. Data are presented as the mean ± SE. Multiple differences among means were evaluated using Duncan’s multiple range tests at a 5% probability level.

## 5. Conclusions

Mild to moderate chilling stress represents a frequently encountered environmental stress for tropical crops such as banana. This study provides thorough comparative molecular analyses on the responses of the LT-sensitive tropical banana genotype to mild and moderate chilling stress. We show that molecular responses of CS banana to mild chilling substantially differ from the response to moderate chilling stress, while banana maintained relatively high net photosynthetic efficiency via upregulation of carbon fixation. Our findings show that banana’s responses to chilling are temperature dependent, which has to be considered within future breeding programs.

## Figures and Tables

**Figure 1 ijms-21-09326-f001:**
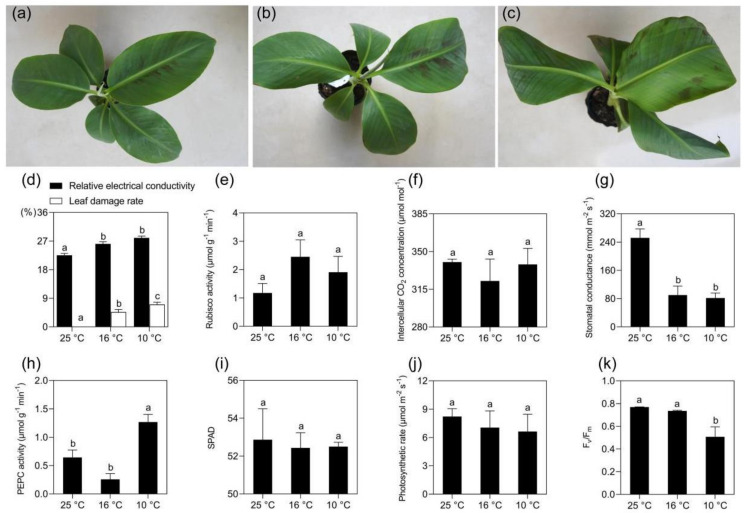
The symptoms and changes in physiological parameters of *Musa* spp. AAA cv. Baxijiao upon chilling stress. (**a**–**c**) Banana plants incubated at 25 °C, 16 °C or 10 °C for 2 days; (**d**–**k**) Determination of the physiological parameters in the *Musa* spp. AAA cv. Baxijiao at 25 °C, 16 °C, and 10 °C. Data represent an average of three replicates ± standard error. Different letters above the columns represent significant difference using Duncan’s multiple range test at *p* < 0.05 after angular transformation of the data.

**Figure 2 ijms-21-09326-f002:**
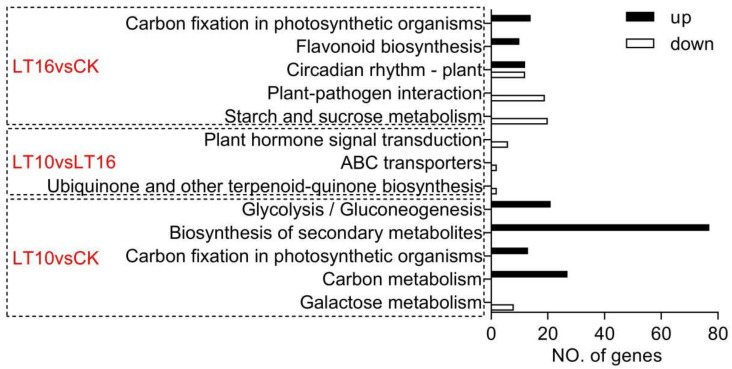
Kyoto Encyclopedia of Genes and Genomes (KEGG) pathways representation for banana (*Musa* spp. AAA cv. Baxijiao) upon mild and moderate chilling stress (corrected *p* ≤ 0.05). The x-axis indicates the number of a specific category of genes in the main term. The y-axis shows the names of functional categories. CK; the control, LT10; low temperature (LT) of 10 °C, LT16; LT of 16 °C.

**Figure 3 ijms-21-09326-f003:**
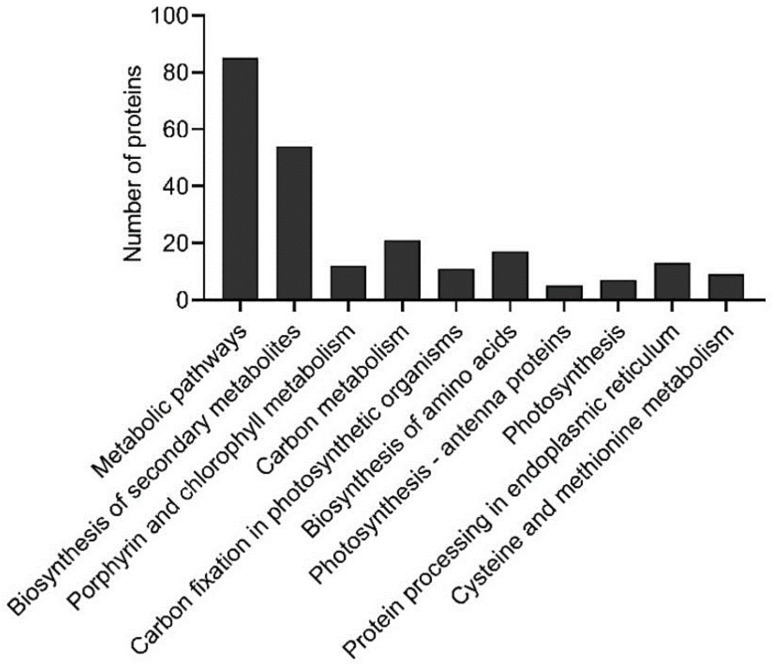
The most enriched KEGG pathways (corrected *p* value < 0.0005) related to differentially abundant proteins in banana (*Musa* spp. AAA) after exposure to 10 °C.

**Figure 4 ijms-21-09326-f004:**
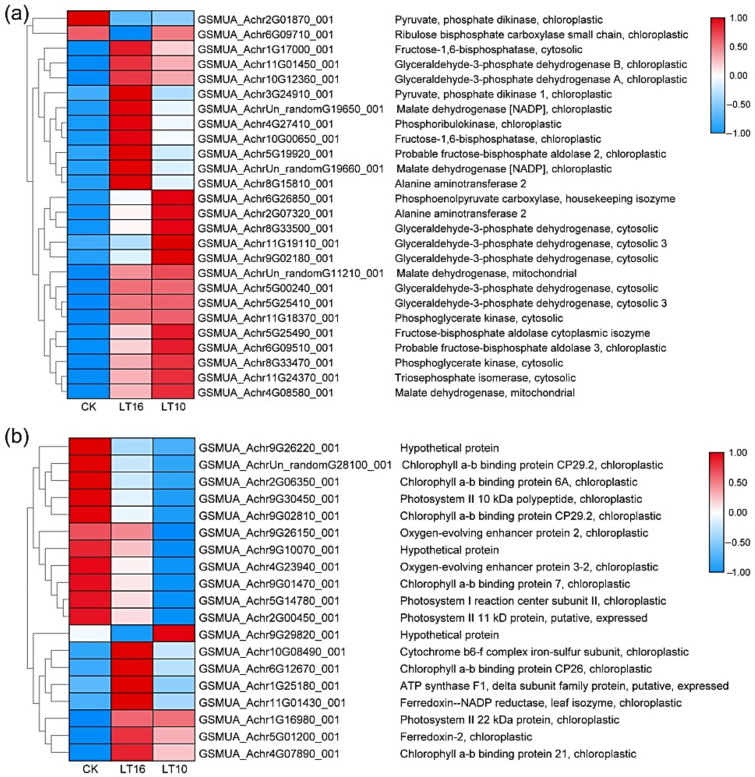
Heatmap showing the changes in expression (expected number of fragments per kilobase of transcript sequence per millions base pairs sequenced) of photosynthesis related genes in banana (*Musa* spp. AAA) in response to chilling stress. (**a**) DEGs related to carbon fixation; (**b**) DEGs related to photosynthesis and photosynthesis-antenna proteins. CK the control, LT10 LT of 10 °C, LT16 LT of 16 °C.

**Figure 5 ijms-21-09326-f005:**
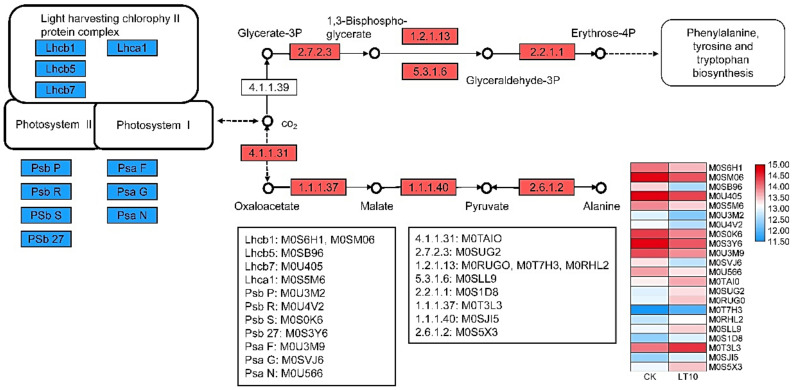
Diagram showing the abundance trends of photosynthesis-related proteins in banana (*Musa* spp. AAA) in response to 10 °C.

**Figure 6 ijms-21-09326-f006:**
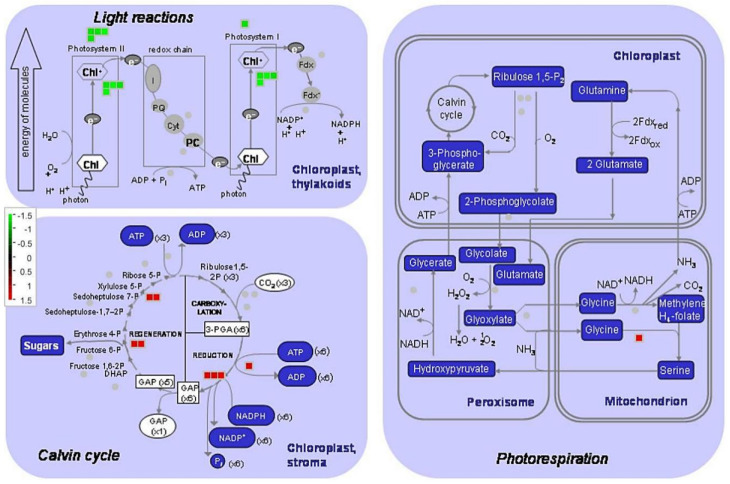
Diagram showing the changes in abundance of photosynthesis-related proteins in banana (*Musa* spp. AAA) in response to a low temperature of 10 °C.

**Figure 7 ijms-21-09326-f007:**
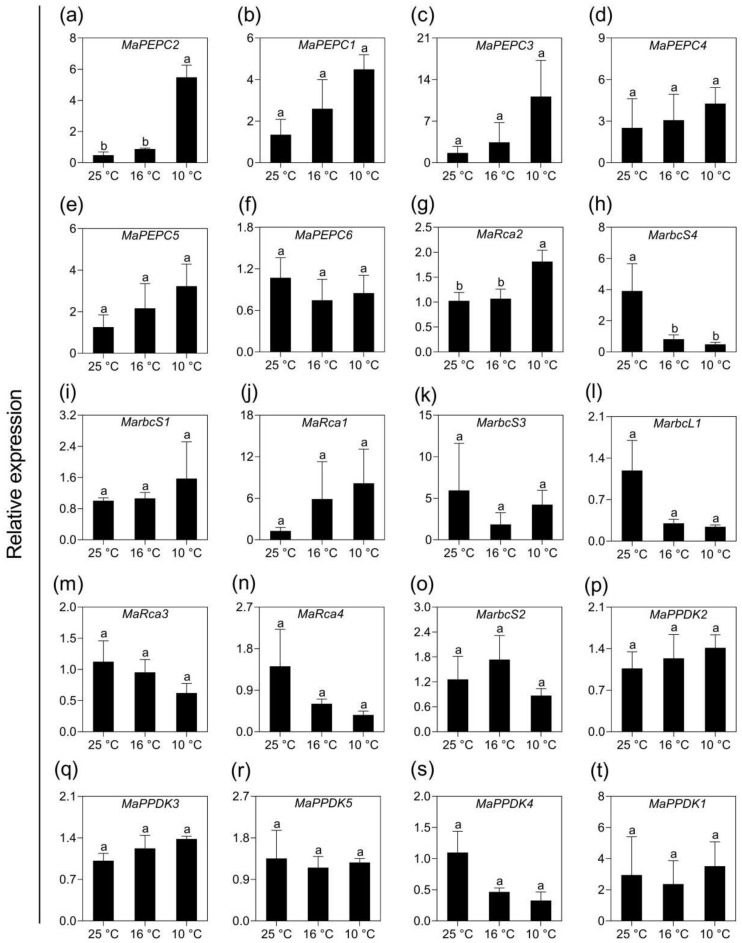
qPCR analysis of the expression of PEPC, Rubisco, and PPDK gene families in banana (*Musa* spp. AAA) exposed to low temperatures. (**a**–**f**) Phosphoenolpyruvate carboxylase (PEPC) genes; (**g**–**o**) Rubisco genes; (**p**–**t**) Pyruvate, phosphate dikinase (PPDK) genes. Data are the average of three replicates ± standard error. Different letters above the columns represent significant difference using Duncan’s multiple range test at *p* < 0.05 after angular transformation of the data.

**Figure 8 ijms-21-09326-f008:**
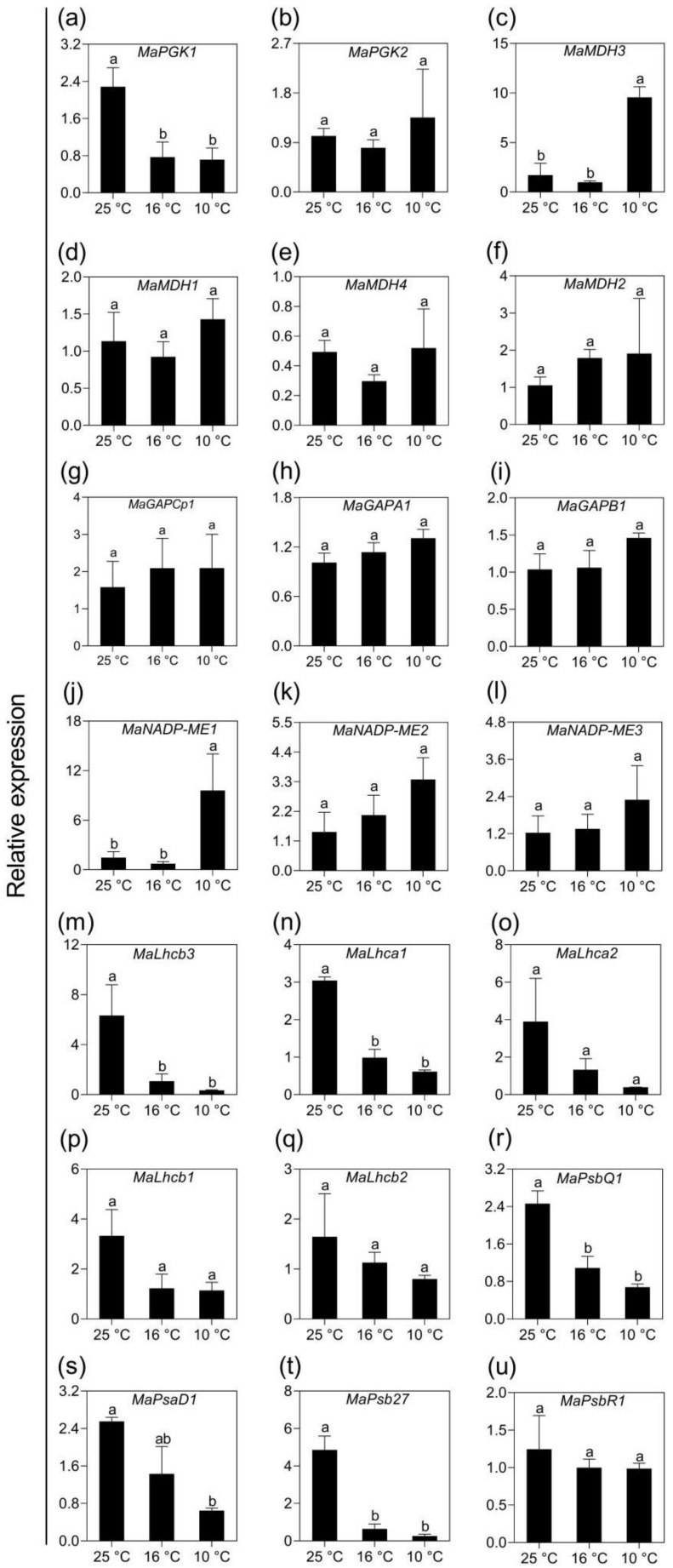
qPCR analysis of the expression of photosynthesis-related genes in banana (*Musa* spp. AAA) exposed to low temperatures. (**a**–**b**) Phosphoglycerate kinase genes; (**c**–**f**) Malate dehydrogenase genes; (**g**–**i**) Glyceraldehyde–3–phosphate dehydrogenase genes; (**j**–**l**) NADP-dependent malic enzyme genes; (**m**–**q**) Light–harvesting chlorophyll protein complex genes; (**r**–**u)** Photosystem I and ll reaction center protein genes. Data represent an average of three replicates ± standard error. Different letters above the columns represent significant difference using Duncan’s multiple range test at *p* < 0.05 after angular transformation of the data.

**Figure 9 ijms-21-09326-f009:**
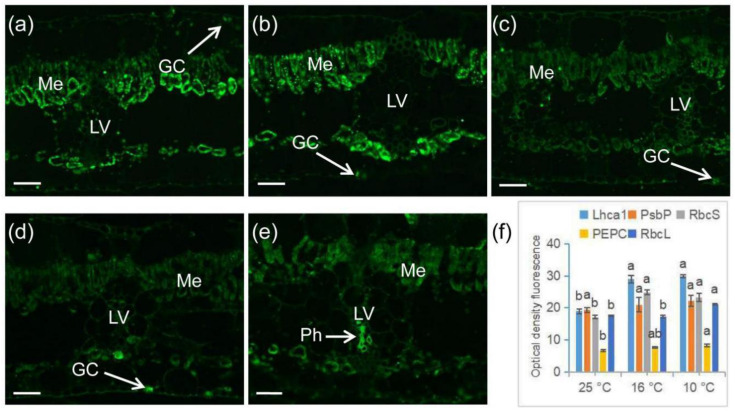
Distribution and localization of proteins involved in photosynthesis and changes in their levels in banana (*Musa* spp. AAA) in response to chilling stress. (**a**) AS01005 antibody recognizing PSI type I chlorophyll a/b-binding protein (Lhca1); (**b**) AS06142-23 antibody recognizing the oxygen evolving complex of PSII (PsbP); (**c**) AS07259 antibody recognizing the Rubisco small subunit (RbcS); (**d**) AS09458 antibody recognizing phosphoenolpyruvate carboxylase (PEPC); (**e**) AS03037 antibody recognizing the Rubisco large subunit (RbcL); (**f**) quantification of fluorescence signal. Data are the average of three replicates ± standard error. Different letters above the columns represent significant difference using Duncan’s multiple range test at *p* < 0.05 after angular transformation of the data (the statistical analysis was carried out for the differences among three temperature points for each antibody separately). GC; guard cells, LV; leaf vein, Me; mesophyll, Ph; phloem. Bars represent 50 µm.

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
