# Peer review of "Acceleration of Carbon Fixation in Chilling-Sensitive Banana under Mild and Moderate Chilling Stresses"

_ijms, 2020, doi:10.3390/ijms21239326_

Round 1

Reviewer 1 Report

The manuscript "Acceleration of Carbon Fixation in Chilling-sensitive Banana under Mild and Moderate Chilling Stresses” studied the responses to mild and moderate chilling stress in banana. The subject is interesting, and the results are good sounded. I would like to thank the authors for their good work and analysis. However, I see the manuscript would be suitable for publication after doing the following minor changes:

  1. I suggest adding figures S4, S7 and S8 to the main figures of the manuscript. Also, please edit the title “4.5.5.
  2. Statistical and Bioinformatic Analysis” as the statistical analysis was already mentioned in separate title.
  3. The conclusion section was missed in the manuscript body. I see that the authors should write a separate conclusion section with more details than mentioned in the abstract section.

Author Response

请参阅附件。

Reviewer 2 Report

The work by Liu et. al. on chilling tolerance in banana put forward interesting hypotheses on cold perception and acclimation in tropical plants. The authors conducted transcriptome and proteome analyses of banana plants exposed to mild and moderate chilling stresses and show that photosynthetic activities are downregulated while the carbon fixation is enhanced upon exposure to cold.

While this work provides some important information on chilling tolerance in a less-studied species such as banana, the manuscript could be improved by incorporating the following changes.

  1. Line 24-25: It requires considerably more biochemical experiments to prove that the cultivar used here is a C3/C4 intermediate. Authors claim this on the basis of immunofluorescence assays performed on chilling-exposed plants. Since more experiments needed to confirm the claim, and the paper do not deal directly with this problem, it is recommended to omit the sentences which claim banana as C3/C4 intermediate in this manuscript. The same comment applies to Lines 255-256
  2. Line 45: Redundancy in the word “productivity”
  3. Line 50: A word may be missing in the part “total of 809 showed”
  4. Figure 1: Legends or explanations are missing for Figure 1 d to k
  5. Tables in the manuscript: Authors write about 9 different tables as the part of manuscript. However, no tables were provided for review. Authors should attach the tables mentioned in the manuscript
  6. Discussion section should include several hypotheses on the differential regulation of photosynthetic and carbon fixation pathways upon cold exposure
